# Reconciling the Biological and Transcriptional Variability of Hepatoblastoma with Its Mutational Uniformity

**DOI:** 10.3390/cancers13091996

**Published:** 2021-04-21

**Authors:** Edward V. Prochownik

**Affiliations:** 1Division of Hematology/Oncology, UPMC Children’s Hospital of Pittsburgh, Pittsburgh, PA 15224, USA; procev@chp.edu; 2The Department of Microbiology and Molecular Genetics, The University of Pittsburgh School of Medicine, Pittsburgh, PA 15224, USA; 3The University of Pittsburgh Hillman Cancer Center, Pittsburgh, PA 15232, USA; 4The University of Pittsburgh Liver Research Center, Pittsburgh, PA 15213, USA

**Keywords:** β-catenin, hepatocellular carcinoma, Hippo pathway, liver cancer, NFE2L2, NRF2, plasminogen activator inhibitor-1 (PAI-1), Serpin E1

## Abstract

**Simple Summary:**

Hepatoblastoma (HB), the most common form of childhood liver cancer, is associated with dual mutation and/or dysregulation of the Wnt/β-catenin and Hippo pathways in ~50% of cases. However, this mutational simplicity cannot explain HB’s biological and histologic diversity. This discussion focuses upon recent work showing that specific β-catenin mutants are key determinants of this HB variability as well as their metabolic and transcriptional signatures. Dysregulation of the anti-oxidant NFE2L2 pathway also contributes to tumorigenesis by being directly transforming in association with either of the other two factors. The transcriptional overlap of tumors generated by pairs of factors identifies crucial targets that likely mediate HB tumorigenesis, behavior and appearance.

**Abstract:**

Hepatoblastoma (HB), the most common childhood liver cancer, is associated with seven distinct histologic subtypes and variable degrees of clinical aggressiveness and presentation. Yet it is among the least genomically altered tumors known, with about half of HBs showing mutation and/or dysregulation of the Wnt/β-catenin and Hippo pathways. This raises the question of how this mutational simplicity can generate such biological and histologic complexity. Recent work shows that the identity of the underlying β-catenin mutation is a major contributor. Mutation or over-expression of the NFE2L2/NRF2 transcription factor, previously thought only to promote anti-oxidant responses, has also recently been shown to accelerate the growth of HBs generated by mutations in the Wnt/β-catenin and Hippo pathways while imparting novel features such as the tumor-associated cysts and necrosis. Moreover, patient-associated NFE2L2 mutations are overtly transforming when co-expressed with either mutant β-catenin or a Hippo pathway effector. The finding that tumorigenesis can be driven by any two arms of the β-catenin/Hippo/NFE2L2 axis has permitted the identification of a small subset of coordinately regulated tumor-specific transcripts, some of whose levels correlate with inferior long-term outcomes in HB and other cancers. Collectively, these findings begin to provide for more refined and molecularly based classification, survival algorithms and design of chemotherapeutic regimens.

## 1. Introduction

Hepatoblastoma (HB) is a rare pediatric tumor that nonetheless constitutes the most common primary liver malignancy of childhood [1,2,3,4,5]. With an incidence of about 1–2 cases/million, only about 50–70 new cases are reported each year in the United States with a similar incidence in the European Union [6,7,8]. Virtually all HBs occur in children <4–5 years of age and the majority appear before 18 months [2,3,5]. Overall survival is approximately 70%, but this is highly dependent upon disease stage and histologic subtype at the time of presentation as well as other features such as patient age and level of serum α-fetoprotein [1,3,4]. Localized Stage I disease is highly curable with surgery alone, whereas survival of those with metastatic or recurrent disease is considerably worse, despite the employment of the most aggressive regimens of combined modality therapy [1,3,4,5].

While the vast majority of HB arises sporadically, several predisposing factors have been identified and include prematurity, Beckwith–Wiedemann syndrome and familial adenomatous polyposis (FAP), with disease incidence in the latter group being as much as 5000-fold higher. The increased diagnosis of HB over the past 2–3 decades likely reflects the improved survival of low-birth-weight infants [1,4,9,10].

At least seven distinct histologic subtypes of HB have been identified with some of these being associated with differences in long-term survival (Table 1) [5,11]. Further complicating this is that different regions of the same tumor may display different histologic features and, in some cases, may resemble hepatocellular carcinoma (HCC) [5].

Most pediatric tumors are associated with far fewer driver mutations than adult tumors, and HBs are the least mutationally diverse of all [2,12,13]. However, approximately 70% of HBs are associated with mutations in the β-catenin transcription factor and a somewhat smaller fraction (~50–60%) of tumors show evidence for dysregulation of the Hippo signaling pathway [2,14,15]. This raises the question of how it is possible to achieve such clinical and histopathologic diversity given the apparent lack of driver mutation variability. A second question is, since no more than about half of HBs deregulate both of these pathways concurrently, what are the drivers in the remaining HBs? In this review, I discuss recent work that has addressed these apparent paradoxes.

## 2. The Molecular Underpinnings of HB

The β-catenin mutations associated with HB, as well as numerous other human cancer types, are highly variable, ranging from missense point substitutions to in-frame deletions as large as 90 amino acids [2,16,17,18,19,20]. This is somewhat unusual given that most oncoprotein mutations are more restricted and predictable by virtue of their being confined to a small number of sites such as codons 12 or 13 in KRAS and codon 600 in BRAF [21,22,23]. The common and most-studied functional consequence of these heterogeneous β-catenin mutations is to abrogate the normal Wnt-responsive cytoplasmic association between β-catenin and the APC complex such that β-catenin is rendered Wnt-independent and stabilized rather than being phosphorylated by APC and rapidly degraded by the proteasome [2,16,17,20,24]. This allows β-catenin to translocate to the nucleus where it interacts with members of the Tcf/Lef family of transcription factors and activates a suite of genes that drive proliferation such as cyclin D and c-Myc [2,18,25]. The increased risk of HB in individuals with FAP [26] derives from the inheritance of germ-line mutations of *APC* that prevent its fully engaging with wild-type (WT) β-catenin in its normal cytoplasmic location. Thus, the primary oncogenic determinant of β-catenin appears to be not whether it is mutated but whether it can be stabilized and constitutively localized to the nucleus in a Wnt-independent manner [17,19,24,27]. Indeed, individuals with FAP retain expression of WT, nuclearly localized β-catenin and, as adults, develop colorectal cancer (CRC) with a frequency approaching 100% regardless of whether or not they develop HB as children [26]. Additionally, the majority of sporadic CRCs contain acquired APC mutations, thus providing molecular links between the sporadic and familial forms of HB and CRC [28].

The precise nature of Hippo pathway dysregulation in HB remains obscure, but its common denominator appears to involve an inability to phosphorylate and degrade the Hippo effector YAP (yes-associated protein) in the cytoplasm [25,29]. Echoing the β-catenin theme, stabilized YAP translocates to the nucleus where it interacts with members of the TEAD transcription factor family and co-activates target genes such as Cyr61, Jag1 and Survivin while also cross-talking with the β-catenin/Tcf complex and co-regulating a subset of its targets [18,25,29]. A key discovery was made in 2014 when Tao et al. [30] demonstrated that hydrodynamic tail vein injection (HDTVI) into mice of Sleeping Beauty vectors encoding the patient-derived β-catenin deletion mutant “Δ(90)” and the nuclearly localized YAP mutant “YAP^S127A^” efficiently induced the rapid growth of tumors that histologically resembled the so-called crowded fetal (CF) HB subtype.

## 3. Accounting for HB’s Clinical and Histopathological Variability

The β-catenin mutational heterogeneity among human HBs [2,16,17,18,19,20] prompted Zhang et al. [20] to ask whether this might account for their highly variable clinical behaviors and histologies. They generated 13 different patient-derived mutants of β-catenin, including Δ(90), and introduced these along with YAP^S127A^ into the livers of recipient mice using HDTVI. They also introduced WT β-catenin to test the hypothesis that it was also potentially transforming so long as sufficient levels were present to overwhelm APC’s capacity for maintaining cytoplasmic confinement. To test whether all β-catenin mutants were oncogenic, this collection included several that, while associated with other cancers, had never been previously described in HB.

Zhang et al. made several pertinent observations that addressed the above questions [20]. First, highly efficient tumorigenesis was observed in all groups, indicating that, at least for the mutations tested, the tumorigenicity of β-catenin mutants was not tissue-restricted. Second, tumors grew at distinct rates, thereby allowing the 14 different groups to be classified into three categories with median survivals of 87.4 ± 6.9 days (Group 1), 124.1 ± 6.1 days (Group 2) and 210.2 ± 17.7 days (Group 3). Third, even small differences in β-catenin mutant identities could be associated with large phenotypic differences, as was seen with missense mutations within the same codon. For example, tumors generated by the S33A and S33Y mutants were associated with Groups 1 and 3, respectively, as were those generated by the S45A and S45P mutants. From these findings, Zhang et al. [20] concluded that β-catenin mutational variability was at least partially responsible for the differential growth rates of human HBs. Finally, WT β-catenin was also oncogenic despite being expressed at relatively low levels relative to endogenous β-catenin.

β-catenin mutational status also markedly impacted tumor histology in ways that were independent of growth rate. For example, rapidly growing Group I tumors arising in the S45A and Δ(90) mutant backgrounds showed the CF appearance typical of the most common human HB subtype [2,3,5,20,31]. In contrast, tumors arising in response to WT β-catenin (Group III) and the K335I point mutant (Group I) more closely resembled HCC, with the former containing scattered areas with CF-like HB appearance and the latter more closely resembling the so-called blastemal-like subtype [5,11]. Group II tumors appeared to be the most variable, ranging from those with predominant HCC-like histology (generated by the T41A mutant) to those with a mixed histology comprised of macrotrabecular, pleomorphic fetal and blastemal elements (generated by the S45P mutant). While the number of mutants studied was too small to allow for general conclusions linking tumor growth rates to particular subtypes, these findings did underscore the general idea that particular histologic subtypes and tumor growth rates were recurrent and a predictable outcome of the aberrant expression of certain β-catenin mutants.

## 4. Different β-Catenin Mutants Are Associated with Differential Subcellular Localization and Biochemical and Molecular Behaviors Features

Zhang et al. [20] next explored whether differential tumor growth rates and appearances were attributable to altered behaviors of the β-catenin mutations. Among the non-mutually exclusive possibilities they considered were changes in β-catenin’s stability, which could be intrinsic in nature or due to its residual association with the APC complex that still permitted some variable degree of β-catenin phosphorylation and proteasome-mediated degradation [14,24,32]. They also considered different efficiencies of β-catenin’s nuclear translocation, its transcriptional activation potential and its gene expression profiles. Zhang et al. [20] found large differences in the absolute levels of mutant β-catenin expression. As expected for a protein that remained fully capable of interacting with and being phosphorylated by APC, WT β-catenin was expressed at low levels and tumors grew slowly, as mentioned above. On the other hand, the S33A and K335I β-catenin mutants were also expressed at low levels yet generated Group I tumors. In keeping with this theme, tumors expressing the highest β-catenin levels, such as S33Y, R582W and Δ(45–58) often grew at intermediate or even slow rates (Groups II and III). Thus, wide variations in each mutant’s stability were observed, but correlated poorly with tumor growth rates and not at all with histology.

In light of the above findings, Zhang et al. [20] examined several other properties of the different β-catenin mutants. Subcellular fractionation studies showed that the efficiency of nuclear localization ranged from >95% in the case of Δ(45–58) to only about 40% for S37A and Δ(20–32). Transient transfections performed in HEK293 cells to assess each mutant’s ability to activate a reporter vector bearing tandem repeats of a β-catenin/Tcf4 binding site in its promoter showed wide variations in response, with WT β-catenin and R582W being ~tenfold less efficient than the most active mutant, Δ(36–53). Yet, despite these widely differing behaviors, all three β-catenin variants generated slowly growing tumors. Notably, tumors generated by WT β-catenin and Δ(36–53) expressed similarly low levels of their respective β-catenin protein and localized about equal fractions of their proteins to the nucleus, despite demonstrating the largest differences in reporter activation. Taken together, these results suggested that the relationships among β-catenin’s subcellular location, inherent transactivation potential and influence over tumor growth rates and histology were complex and probably multi-factoral.

Another possible explanation for some of the above differences arose from the observation that the slowly growing neoplasms generated by WT β-catenin actually grew at the same rate as Group I tumors once they were established but showed a much longer lag phase. This suggested the possibility that only a subset of transduced hepatocytes was susceptible to transformation by WT β-catenin and further suggested that secondary events were necessary to initiate tumor growth. Zhang et al. [20] tested this by delivering a co-equal mix of WT β-catenin and Δ(90). Because each β-catenin was fused to different epitope tags, they could be readily distinguished. These tumors appeared rapidly, thereby indicating that they were driven primarily, if not exclusively, by Δ(90). Moreover, the vast majority of WT β-catenin in these tumors was now found in the cytoplasm. This provided compelling support for the idea that tumors arising in response to WT β-catenin dysregulation originate in a subset of transfected cells that allow for its higher nuclear concentrations.

Nine of the 14 tumor groups were examined by RNAseq for gene expression differences, which ranged from 6820 to 4222 relative to control livers. Inter-tumor group differences ranged from as many as 1515 to as few as only one. Functional categorization of the differentially expressed transcripts among the various tumors groups indicated that the most generally impacted pathways were those pertaining to peroxisomal-based ω-fatty acid oxidation (FAO), lipid and xenobiotic metabolism and to several mitochondrial-based metabolic pathways including the TCA cycle, β-FAO and branched chain amino acid oxidation. Examination of a previously described 613 member set of direct Tcf4/β-catenin target genes identified by ChIP-seq in primary murine hepatocytes [33] showed an average of 53% of these to be deregulated in tumors relative to livers (range 44.4–61.0%). Collectively, these results indicated that tumors arising as a consequence of the enforced expression of different β-catenin mutants dysregulate distinct but substantially overlapping members of a large gene set, many of which are direct targets. Each tumor group’s phenotypes and behaviors thus appear to reflect the integrated expression of its specific gene expression profile.

Among β-catenin’s most prominent direct targets is the *myc* gene, whose encoded protein, Myc, is a potent transcription factor in its own right and whose dysregulation in HB initiates its own separate transcriptional cascade [18,34,35,36]. Conditional inactivation of *myc* in murine hepatocytes markedly impairs HB growth and alters other behaviors in response to the enforced co-expression of Δ(90) and YAP^S127A^ [35,36,37]. Zhang et al. [20] demonstrated that 32.2–43.6% of a >600 member set of direct Myc target genes were dysregulated in response to the above nine WT and mutant β-catenin proteins, thus likely identifying additional factors that contribute to tumor heterogeneity. Myc itself, normally expressed at barely detectable levels in liver [38], was also variably up-regulated by tumors in each of the above groups. L-Myc, which is expressed at high levels in livers, is much less active than Myc as a transcriptional activator and imparts different phenotypes [35,36,39,40], was variably down-regulated in tumors.

Collectively, the findings of Zhang et al. [20] indicate that the underlying β-catenin mutation’s identity plays a preeminent role in determining many of the features of HB, including, most notably, the rate of tumor growth, the histopathologic appearance of the tumor and its transcriptional repertoire. These findings further suggest that knowledge of the β-catenin mutant’s identity might be used to gauge tumor aggressiveness and potential for recurrence and might aid in the design of more personalized therapies. Lastly, the findings underscored the significant degree to which seemingly subtle mutational differences, even within the same codon, can influence tumor behavior in profound ways. Similar examples of the impact of seemingly minor mutational differences and their relationship to chemotherapy response and/or survival have been described for the epidermal growth factor receptor in non-small-cell lung cancer, KRAS in CRC and PIK3CA and MAP2K(MEK1) in other cancers [16,41,42,43,44,45,46,47].

## 5. HB Pathogenesis May Involve Culprits Other than β-Catenin and the Hippo Pathway

Given that ~80% of HBs harbor β-catenin mutations and a somewhat smaller fraction deregulate the Hippo pathway [2,3,14,18,30,34], it remains the case that at least half of HBs do not possess this pro-oncogenic combination. Cases of HB with FAP-associated APC mutations or mutations in other APC Complex components such as Axin1 and Axin2 are exceedingly rare and likely account for <2–5% of these remaining tumors [28]. Thus, other pathways must exist to explain this gap.

Wang et al. [48] recently investigated the role of the oxidant-responsive transcription factor NFE2L2/NRF2 (NFE2L2) in the pathogenesis of this large non-canonical subset of HBs. Again following a theme that recalls the regulation of β-catenin and YAP, NFE2L2 is normally maintained in an inactive cytoplasm complex with Kelch-like ECH-associated protein 1 (KEAP1), which mediates NFE2L2’s rapid proteasome-mediated decay so as to ensure its low-level basal expression and transcriptional quiescence [49]. In response to oxidative or electrophilic stress, however, multiple KEAP1 cysteine residues are oxidized, causing the release, stabilization and nuclear translocation of NFE2L2, which, upon interacting with members of the small bZIP Maf family, alters the expression of several hundred target genes and mediates an anti-oxidant response [49,50,51]. An average of 4.6% of HBs, (range 0–9.8%) harbor missense mutations in one of NFE2L2’s two KEAP1-interacting domains, which leads to NFE2L2’s stabilization and nuclear localization even in the absence of any exogenous stress [19,48,52,53,54]. Additionally, as many as 20% of certain adult epithelial neoplasms and 50% of HBs amplify the WT *NFE2L2* gene at 2q32.1 and allow its encoded protein to stoichiometrically override the KEAP1 checkpoint [19,48,55,56]. It has been proposed that, depending on when during tumorigenesis NFE2L2 dysregulation occurs, it can either suppress or facilitate tumorigenesis. For example, at early times, NFE2L2 activation can protect against oxidant-mediated DNA damage and the accumulation of oncogenic lesions and thus serve as a tumor suppressor [49,50,51]. At later times, NFE2L2 over-expression can increase the tolerance for high levels of oncogene-mediated reactive oxygen species, thus allowing for previously unachievable levels of oncoprotein expression and more robust aberrant signaling [51,52,53,55,57].

Employing the above murine HB model, Wang et al. showed that the enforced co-expression of Δ(90) and YAP^S127A^, and either of two patient-derived point-mutant forms of NFE2L2 (L30P or R34P [L30P/R34P]), but not WT NFE2L2, generated HBs that grew significantly more rapidly than Group I tumors [20,48]. Additionally, these tumors contained large and widespread areas of necrosis and innumerable fluid-filled cysts, neither of which had been observed in the tumors previously described by Zhang et al. or others [2,20,30,35,36], although cysts are occasionally observed in human HBs (Figure 1) [58]. Consistent with the anti-oxidant function of NFE2L2 [49,50,51], tumor cells expressing either L30P or R34P and exposed briefly to H_2_O_2_ in vitro recovered from the oxidative stress much more rapidly than either control Δ(90) + YAP^S127A^ tumors or those co-expressing WT NFE2L2 [48]. A similar acceleration of growth by either L30P or R34P was seen in conjunction with the R582W β-catenin mutation that generated Group III tumors [20]. These L30P/R34P-expressing tumors now grew about as rapidly as Group I tumors and were also associated with cysts and necrosis, thus arguing that these features were not a consequence of an excessive growth rate that allowed the tumors to outstrip their vascular supply. Rather, cyst formation seemed to represent an intrinsic property of the tumors imparted by the specific co-expression of L30P/R34P.

Unexpectedly, Wang et al. [48] also found that any two combinations of Δ(90) β-catenin, YAP^S127A^ and L30P or R34P were tumorigenic, although the entire triad was needed to promote cystogenesis, necrosis and accelerated growth. This indicated that, rather than simply facilitating tumor growth by creating a more conducive environment [50,51,55], NFE2L2 mutants were directly oncogenic in their own right and equal to that of either Δ(90) or YAP^S127A^ when expressed in a pair-wise manner. This finding provided a feasible and testable explanation for the high rates of *NFE2L2* gene mutation or copy number variation in HBs and other cancers by suggesting that HBs lacking one of the signature β-catenin or Hippo pathway lesions might instead rely on NFE2L2 dysregulation.

Unsurprisingly, tumors generated by either the full complement of Δ(90) YAP^S127A^ and L30P/R34P mutants, or any two combinations of the three oncoproteins, possessed distinct metabolic, biochemical and molecular signatures [48]. In the latter case, each of the five possible tumor groups differed from one another by 821–4676 transcripts. Surprisingly, some of these tumors thus differ from one another to a greater degree than do some mutant β-catenin, +YAP^S127A^-generated HBs and control livers [35,46,48]. These distinct gene expression repertoires, with limited amounts of overlap among groups, allowed Wang et al. to identify transcripts specifically regulated by each oncoprotein as well as a common subset of 41 so-called “BYN” transcripts shared among all tumor cohorts and postulated to participate in a common transformation pathway [48].

Wang et al. [48] made several sassumptions in further refining the BYN subset to its most basic core elements. First, they eliminated 12 transcripts that were not always deregulated in the same direction in all tumor groups. An additional seven were eliminated as they were not deregulated in all tumors generated from the β-catenin mutants studied by Zhang et al. [20] or in an independently derived set of slowly growing Δ(90) + YAP^S127A^ HBs generated in mice with hepatocyte-specific knockout of *myc* and/or the *myc*-related transcription factor *chrebp* [35,36]. The remaining 22 transcripts were thus posited to encode a common set of factors mediating transformation but having little relevance to tumor initiation or growth rates, histology or specific metabolic signatures.

Several features of the above 22 transcripts supported the notion that they were highly relevant to HB pathogenesis. First, ten transcripts perfectly identified a high-risk category of human HBs that had been previously stratified by two other groups based on expression differences of 16 or four transcript sets [43,56]. None of the three gene sets contained any common elements, thereby suggesting that the combined group of 30 transcripts might be particularly useful for HB prognostication. Second, using data from The Cancer Genome Atlas, Wang et al. showed that 17 of the 22 transcripts correlated with survival in 14 adult cancer types and that, in some cases, these correlations involved multiple cancers [48]. Finally, 16 of the transcripts showed associations with at least one of the ten known Cancer Hallmarks [59] and eight were associated with between five and nine. By way of comparison, the potent oncogenes *KRAS* and *PIK3CA* and the tumor suppressor *TP53* are associated with five, six and eight of these Hallmarks, respectively. Collectively then, multiple HB-associated BYN transcripts were found to correlate quite strongly with survival not only in HB but in multiple other cancers.

## 6. The Beginning of a Molecular Reconstruction of HB

The findings summarized above should improve our understanding of the molecular pathogenesis of HB by shifting the causative focus from the Wnt/β-catenin and the Hippo pathways in general to a small subset of β-catenin-, Hippo pathway- and NFE2L2-regulated target genes that are widely deregulated across multiple human cancers as well as HB. Toward this end, Wang et al. [48] noted that one of the above BYN group gene members, namely *serpine1*, was not only the second most highly up-regulated (43.7-fold relative to liver) but was also the only gene whose human ortholog contained multiple Tcf/Lef (β-catenin), TEAD (YAP) and ARE (NFE2L2) elements in its proximal promoter. Serpin E1 protein levels were also highly elevated in Δ90) + YAP^S127A^ + L30P/R34P tumors and both plasma and cyst fluid from tumor-bearing mice contained increased levels of serpin E1.

Although classically associated with fibrinolysis, serpin E1, also known as plasminogen activator inhibitor 1, plays numerous ancillary roles in tumor growth, matrix remodeling, angiogenesis and metastasis [60]. It was also one of the transcripts mentioned in the above section whose deregulation was highly predictive of survival in HB and six different adult cancer types [48].

Interestingly, elevated plasma levels of serpin E1 have been reported in patients with polycystic ovary syndrome, and ovary-specific serpin E1 over-expression in mice is highly cystogenic [61,62]. Therefore, hypothesizing that serpin E1 might be able to at least partially substitute for L30P/R34P, Wang et al. [48] found that its co-expression with Δ(90) and YAP^S127A^ did not alter tumor growth rate or induce cysts but did promote widespread necrosis. Wang et al. [48] therefore hypothesized that serpin E is likely to be only one of several (if not all) BYN members that cross-talk and cooperate to impart both common and unique features to different tumor cohorts.

## 7. Remaining Questions

The recent work of Zhang et al. and Wang et al. [20,48] discussed here raises a number of additional points and questions that should now be readily addressable. For example, it suggests that the ultimate behavior, appearance and transcriptional environment of any given HB are likely to be a consequence of the integrated transcriptional output of at least two of three pathways that are deregulated by the oncogenic activation of Wnt/β-catenin, Hippo and NFE2L2 signaling. Since all three factors control distinct but overlapping transcription programs and since the output of the β-catenin pathway is highly mutant-dependent, might it be possible that additional subtleties in tumor appearance and behavior are mediated through equally subtle differences in the nature of the defects affecting Hippo and NFE2L2 signaling? Are certain β-catenin mutations, certain pro-oncogenic defects in the Hippo signaling pathway and certain NFE2L2 mutations more or less likely to be associated with one another and, if so, why? Can any of the three gene sets now having been described as being predictive of HB patient survival [34,43,48] be used collectively or in a more sophisticated way to permit increased prognostic power and/or therapeutic design? Finally, how do individual members of the BYN transcript family cooperate with one another to recapitulate any or all of the features of HB?

Sekiguchi et al. recently employed a multi-omics approach to classify 59 HBs [63], and showed that epigenetic changes are not only widespread but may also contribute to heterogeneous HB appearances and clinical behaviors. 56 of the tumors contained β-catenin mutations and an additional four contained *APC* gene mutations (i.e., a 100% incidence of Wnt/β-catenin/APC pathway dysregulation). While no mention was made regarding Hippo or NFE2L2 pathway deregulation, 31% of the tumors showed whole-arm amplification of chromosome 2q, which encodes *NFE2L2*. Interestingly, Sekiguchi et al. identified three distinct tumor cohorts groups (E1, E2 and F) based upon the genome-wide methylation patterns of 39 of the tumors, with the E1 and E2 groups being associated with an inferior overall prognosis. The most differentially expressed hypomethylated gene in these cohorts was *NQO1*, a common and well-characterized NFE2L2 target [64]. Although difficult to correlate with our own previously reported findings [20,48], these recent results do raise the possibility that different β-catenin mutations, perhaps in combination with differential dysregulation of the Hippo and/or NFE2L2 pathways, might alter the expression of the various methylases and demethylases that collectively regulate the HB transcriptome. Similarly altered expression of micro-RNAs and long, non-coding RNAs could conceivably contribute to HB gene expression profiles post-transcriptionally as well [65,66].

## 8. Conclusions

The work summarized in this review addresses several long-standing questions that have confronted those who study HB and its pathogenesis. Among the most pressing is how to explain the well-known histologic diversity of this genetically monotonous cancer. It is now apparent that even subtle variations in β-catenin mutations, indeed even those involving the same codon, can exert significant influence over tumor growth rates, histology and transcriptional profiles, with each mutation regulating both common and unique subsets of target genes in ways that are complex and still poorly understood [48]. Another question is the nature of the underlying oncogenic underpinnings of the ~50% of HBs lacking concurrent mutation/dysregulation of both β-catenin and Hippo pathways. NFE2L2, a transcription factor long been thought to initiate a broad range of oxidative and electrophilic stress responses, is now known to be directly oncogenic and to cooperate with either or both of the other two pathways to initiate de novo tumorigenesis, accelerate its progression, alter its metabolic and transcriptional features and impart new features. Finally, the large number of genes that are deregulated in each of these individual tumor cohorts relative to the normal liver has made it difficult to separate those that are the most critical for transformation from those with either ancillary roles or none at all. The identification of a small common subset of so-called BYN genes from the several thousand that are deregulated in any given tumor provides a readily manageable number that avail themselves to more in-depth scrutiny in future studies. The fact that the expression of virtually all the genes in this collection correlate with adverse prognosis in HB or other human cancers (or, in some cases, multiple cancers) is testimony to the underlying role(s) they seem likely to play.

## Figures and Tables

**Figure 1 cancers-13-01996-f001:**
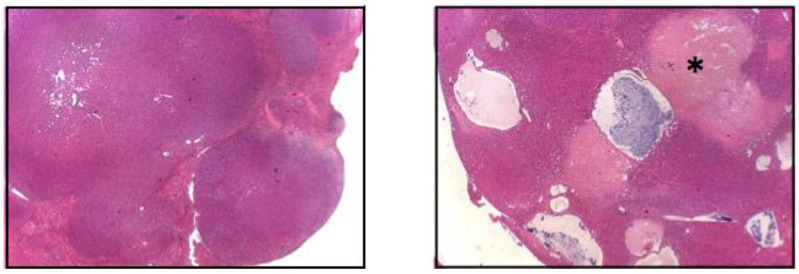
The co-expression of Δ(90) β-catenin, YAP^S127A^ and L30P/R34P induces HBs with widespread cyst formation and necrosis. Low power (5×) images of H&E stained tumor HBs. The (**left panel**) shows the typical crowded fetal-like appearance associated with the co-expression of Δ(90) β-catenin and YAP^S127A^ [20,30,35]. The (**right panel**) shows a tumor generated by the combined over-expression of Δ(90) β-catenin + YAP^S127A^ + R34P. Note the cysts and adjacent regions of widespread necrosis (asterisk). See [48] for additional, higher-power H&E-stained images of the crowded fetal morphology (Reprinted with permission from ref. [48]. Copyright 2021 Children’s Hospital of Pittsburgh of UPMC).

**Table 1 cancers-13-01996-t001:** Histologic classification of hepatoblastoma (adopted from references [3,5]).

Type	Subtype and/or Properties
**Epithelial**	
Fetal	• Well-differentiated: uniform with round nuclei, cords with minimal mitotic activity; may be associated with EMH * • Crowded: mitotically active, prominent nucheoli • Pleomorphic: poorly differentiated, high nuclear:cytoplasmic ratio; moderate anisonucleosis • Anaplastic: marked nuclear enlargement, pleomorphism and hyperchromosia; abnormal mitoses
Embryonal	Smaller cells; high nuclear:cytoplasmic ratio; primitive tubules may be associated with EMH
Macrotrabecular	Fetal or embryonal type growing in clusters of 5 or more cells between sinusoids
Small cell undifferentiated	About half the size of fetal HB; minimal pale & amphophillic cytoplasm; oval/round nuclei; fine chromatin; inconspicuous nucleoli
Cholangioblastic	Presence of bile ducts that may be quite prominent
**Mixed**	
Stromal derivative	Spindle cells/blastema; presence of other components (osteoid, cartilage, muscle)
Teratoid	Mixed; primitive endoderm; presence of neural derivatives; squamous and/or glandular components, melanin

* EMH: extramedullary hematopoiesis.

## Data Availability

Access to all relevant data from the author’s laboratory can be obtained from [20,48].

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
