# Peer review of "Reconciling the Biological and Transcriptional Variability of Hepatoblastoma with Its Mutational Uniformity"

_cancers, 2021, doi:10.3390/cancers13091996_

Round 1

Reviewer 1 Report

In this review, Dr. Prochownik summarize recent experimental findings that could explain part of the transcriptional variability of hepatoblastoma (HB), a tumour with an extremely low mutational burden (<2 mutations/Mb) and with a major mutated oncogene (CTNNB1).  In particular, the manuscript is focused on explaining two recent experimental studies of HB tumorigenesis in mice of the Dr. Prochownik’s lab using CTNNB1, YAP and NFE2L2 mutations (references 20 and 48).

The review is very well written and it includes major references in the HB field; however, it be necessary to include some comments about the following papers:

  • Cairo et al (Cancer Cell, 2008; PMID: 19061838), Carrillo-Reixach et al (J Hepatol, 2020; PMID: 32240714) and Sekiguchi et al (NPJ Precission Oncology, 2020: PMID: 32656360) as references of papers that describe the transcriptomic variability of HB. In particular, Cairo et al reported the different expression of MYC and beta-catenin target genes in the C1/C2 HB subclasses. Cairo et al as well as Carrillo-Reixach et al also associate the different transcriptomic variations of HB subclasses with its histology (C1 tumors are associated with mainly tumors with a predominant fetal differentiation), reinforcing the idea that well-known pathological heterogeneity of HB is linked to its transcriptional variability.
  • Page 6, line 245: it is written that “Many tumors, including about 3-5% of HBs harbor missense mutations in one of NFE2L2’s two KEAP1-interacting domains which leads to NFE2L2’s stabilization and nuclear localization even in the absence of any exogenous stress [19,52,53].” However, author only refer to the study of Sumazin et al (reference 19) in which NFE2L2 were found in 3/34 HBs (9%). Please include the reference of Eichenmüller et al (J Hepatol, 2014; PMID: 25135868) in which authors reported mutations of NFE2L2 in 2/15 cases (13%) and correct the percentage of HB with NFE2L2 mutations.
  • Page 6, lines 246-247: author should specify that NFE2L2 is located at 2q32.1 and also refer to Cairo et al (Cancer Cell, 2008; PMID: 19061838) and Carrillo-Reixach et al (J Hepatol, 2020; PMID: 32240714) as papers that together with Sumazin et al reported a gain of 2q in about 50% of HBs.
  • López-Terrada et al (Human Pathol 2009; PMID: 192005799) where authors reported that the Pure Fetal HB is associated with large CTNNB1 deletions.

Author should check the references of the review, since some mistakes have been found:

  • Page 7, line 310: reference 34 is not correct. Author should add in the review the reference Cairo et al (Cancer Cell, 2008; PMID: 19061838) that describes the 16-gene signature.
  • Page 5, line 233: reference 28 does not correspond to the Wang et al paper which is reference 48.

Author Response

Comment 1. Cairo et al (Cancer Cell, 2008; PMID: 19061838), Carrillo-Reixach et al (J Hepatol, 2020; PMID: 32240714) and Sekiguchi et al (NPJ Precission Oncology, 2020: PMID: 32656360) as references of papers that describe the transcriptomic variability of HB….

Response: The Cairo et al and Carillo-Reixach et al papers have now been referenced as suggested by the reviewer. More important perhaps is a new discussion of the Sekiguchi et al paper that had escaped my attention during the original preparation of this manuscript. While not directly pertinent to the b-catenin/YAPS127A/NFE2L2 mutational activation that is the main focus of the review, it is quite important in the sense that it allows an appreciation of the fact that the behavior and appearance of HBs may be also be under epigenetic control and that such control could well be regulated by any or all of the above factors. This is discussed in terms of how b-catenin/YAPS127A/NFE2L2 dysregulation might cooperate to affect the expression of the numerous methylases and demethylases that collaborate to determine the epigenetic balance of the tumor genome. This discussion, along with brief mention of miRNA and lncRNA regulation as well, is now included at the end of Section 7 as one of the “remaining questions” that should be the focus of future work.  

Comment 2: Page 6, line 245: it is written that “Many tumors, including about 3-5% of HBs harbor missense mutations in one of NFE2L2’s two KEAP1-interacting domains which leads to NFE2L2’s stabilization and nuclear localization even in the absence of any exogenous stress [19,52,53].”

Response: The reference of Eichenmuller et al has been added as suggested. This showed the incidence of NFE2L2 point mutations to be 9.8% in an expanded series of 47 primary tumors and 4 HB cell lines. The 3-5% incidence of NFE2L2 point mutations cited in the review is correct, although it was intended to represent the average incidence rather than the range of all the cited sources. In our recent paper (Wang, Cell Mol Gastroenterol Hepatol. 2021 Feb 20:S2352-345X(21)00037-0.), we reported this 4.6% incidence from a compilation of 194 tumors from COSMIC data base and several other references, including the above Eichemuller et al report, which reported the highest incidence. So as to avoid any mis-interpretation, we have modified our review to read “An average of 4.6% of HBs, (range 0-9.8%) harbor missense mutations in one of NFE2L2’s two KEAP1-interacting domains which leads to NFE2L2’s stabilization and nuclear localization even in the absence of any exogenous stress [19,48,52-54].” We would also point out that the work of Sekiguchi et al that the reviewer pointed out (Sekiguchi et al. NPJ Precis Oncol. 2020.4:20.) examined 59 primary HBs using a multi-omics approach. They did not specifically comment upon the identification of any NFE2L2 mutations. Because of this uncertainty, we did not add this cohort to our original compilation of the above 194 NFE2L2 mutations. However, if we were to assume that no mutations were identified, which seems likely, the incidence would then be 3.3% [nine of 253 (194+59): ref. 48]. On the other hand, 30% of the Sekiguchi et al samples were reported to contain whole-arm copy number gains of chromosome 2q, which is consistent with NFE2L2 gene amplification.

Comment 3: Page 6, lines 246-247: author should specify that NFE2L2 is located at 2q32.1…

Response: The chromosomal location of NFE2L2 has now been mentioned and the Cairo et al and Carrillo-Reixach et al references added.

Comment 4: López-Terrada et al (Human Pathol 2009; PMID: 192005799) where authors reported that the Pure Fetal HB is associated with large CTNNB1 deletions.                                                                                                                 

Response: Lopez-Terrada et al, demonstrated that the HepG2 cell line was in fact associated with a large, in-frame deletion of b-catenin. We did not cite this work as it did not include any primary tumors, was done with only a single cell line (the purpose of which was to settle a long-standing debate as to whether HepG2 originated from HB or HCC) and did not establish that b-catenin deletion mutants tended to be seen more commonly in any particular HB subtype (or even with HB for that matter). Subsequent reports by Dr. Lopez-Terrada and her colleagues have placed b-catenin mutations and their relation to HB in a broader context and these references were cited in the review (Czauderna et al. Curr Opin Pediatr. 2014,26:19-28 Ranganathan et al. Pediatr Dev Pathol. 2020,23:79-95.).

Comment 5-7: Author should check the references of the review, since some mistakes have been found:

Response: These errors have been corrected.

Reviewer 2 Report

The author attempts to answer the question of how to reconcile the biological and transcriptional variability of hepatoblastoma (HB) with its mutational homogeneity. In other words how it is possible to achieve a high degree of clinical and histopathologic diversity regarding this rare pediatric tumor, given the apparent lack of mutational variability in the mutation drivers. The author has vast experience in the subject matter of the paper, so he seems most entitled to show a sort of extended research discussion of his own scholarly achievements. The author's two papers (ref. 20, and 48) are primarily discussed, but in relation to other research papers in the subject area. The hypotheses of the review are clear, scientifically justified and do not raise any objections. The review is very important from clinical point of view and may carry practical importance. The literature comprising 60 papers is mostly of recent years, only about 28% are from years below 2010. 

Minor editorial notes:

1.      Table 1 is not very readable. Please improve its technical quality. 2.      Figure 1 is also not very convincing, it seems to me that the magnification of 5x is too low to see the histopathological features present in this HB fragment.Would it be possible to replace at least part of this figure?3.      Please also check if 70% or 80% of HBs are associated with beta-catenin mutations (lines 61 vs. 227)?4.      There is wrong journal numbering next to the name of Wang et al. (line 233), it should be 48 instead of 28.5.      Minor literal errors- line 201 - "myc gene" instead of "mycc gene". "colorectal" instead of "colo-rectal (line 86); "subtype" instead of "subtype" (line 102).6.      Please clean up and check the table of references according to the requirements of the journal, adding doi numbers, etc.

Author Response

Minor editorial notes: 

Comment 1: Table 1 is not very readable. Please improve its technical quality.

Response: The Table has been reconfigured and made more readable. Please note also that the image embedded in the manuscript is a screen shot and that a higher resolution image will be provided prior to publication.

Comment 2: Figure 1 is also not very convincing, it seems to me that the magnification of 5x is too low to see the histopathological features present in this HB fragment.

Response: The purpose of this figure was to emphasize the innumerable cysts and widespread necrosis that accompany virtually all HBs that co-express b-catenin + YAP + NFE2L2. This can only be appreciated with low-power images as presented in Fig. 1. Many higher-power images and in-depth discussions of the histologic differences among various tumor types were presented in our recent publication (Wang et al. Cell Mol Gastroenterol Hepatol. 2021 Feb 20:S2352-345X(21)00037-0.) and the histology of D(90)b-catenin+YAPS127 HBs has been well-described by us and others (Tao et al. Gastroenterology. 2014.147:690-701; Min et al. Am J Pathol. 2019.189:1091-1104; Wang et al. J Biol Chem. 2016. 291:26241-26251; Zhang et al. J Biol Chem. 2019.294:17524-17542.). Mention that numerous other images are available has been added to the Figure legend of the revised manuscript

Comment 3: Please also check if 70% or 80% of HBs are associated with beta-catenin mutations (lines 61 vs. 227)?

Response: Line 227 has been changed so as to be consistent with the conservative 70% incidence cited in line 61. These figures were obtained from various sources in the literature citing mutation incidences of 50-87% in the CTNNB1 gene in human HB (Koch et al. Cancer Res. 1999.59:269-273; Cairo et al. Cancer Cell 2008.14:471-84; Lopez-Terrada et al. Hum Pathol. 2009.40:783-794; Czauderna et al. Curr Opin Pediatr. 2014.26:19-28; Sylvester and Colnot. Gastroenterology. 2014.147:562-565; Eichenmuller et al. J Hepatol. 2014.61:1312-1320. Tao et al. Gastroenterology. 2014.147:690-701; Bell et al. Gene Expr. 2017.17:141-154).

Comment 4: There is wrong journal numbering next to the name of Wang et al. (line 233), it should be 48 instead of 28.

Response: The incorrect reference has been changed

Comment 5: Minor literal errors- line 201 - "myc gene" instead of "mycc gene". "colorectal" instead of "colo-rectal (line 86); "subtype" instead of "subtype" (line 102).    

Response: The minor typos have been corrected in the revised manuscript 

Comment 6: Please clean up and check the table of references according to the requirements of the journal….

Response: All references have now been formatted in the style of the journal, which does not require DOI numbers.